# A Lipoma Arborescens Probably Causing Significant Osteoarthritis of the Elbow in a Young Man

**DOI:** 10.3390/diagnostics15151888

**Published:** 2025-07-28

**Authors:** Won-Jong Bahk, Seungyup Shin, Junho Jang, Kyung Jin Seo, Yongju Kim, Hyunjung Kim

**Affiliations:** 1Department of Orthopedic Surgery, Cheju Halla General Hospital, Jeju 63127, Republic of Korea; wjbahk@gmail.com (W.-J.B.); syshin.md@gmail.com (S.S.); jph319@naver.com (J.J.); 2Department of Hospital Pathology, Uijeongbu St. Mary’s Hospital, College of Medicine, The Catholic University of Korea, Seoul 06591, Republic of Korea; ywacko@catholic.ac.kr; 3Department of Radiology, Uijeongbu St. Mary’s Hospital, College of Medicine, The Catholic University of Korea, Seoul 06591, Republic of Korea; rladydwn1005@gmail.com; 4Department of Laboratory Medicine, College of Medicine, The Catholic University of Korea, Seoul 06591, Republic of Korea

**Keywords:** lipoma arborescens (LA), osteoarthritis, intra-articular lesion, joint, bone

## Abstract

Lipoma arborescens (LA) is a rare, non-neoplastic, intra-articular, mass-like lesion with villous lipomatous proliferation that replaces and distends the synovium, particularly in the knee joint. A few cases have been sporadically reported to affect the shoulder, elbow, wrist, hip, and ankle. The authors would like to present a rare and unique case of LA in the elbow joint with significant osteoarthritis in a 24-year-old young man, which suggests that a longstanding pre-existing LA can give rise to severe degenerative arthritis even in young patients unless diagnosed early and adequately treated.

**Figure 1 diagnostics-15-01888-f001:**
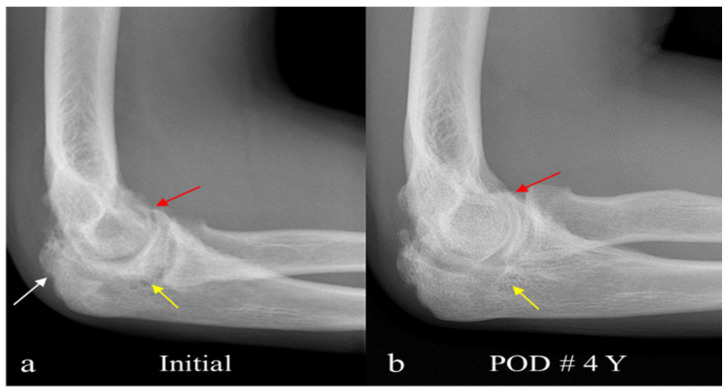
(**a**) Plain radiograph. A 24-year-old man presented with a palpable mass in the antecubital fossa, along with gradually worsening pain and limitation of joint motion in his right elbow over four years. Physical examination revealed a palpable, soft mass deep in the antecubital fossa. Elbow motion was moderately reduced to 30° extension and 100° flexion. Laboratory data, including rheumatoid factor, uric acid, and C-reactive protein levels, were within normal limits. He denied any personal or family history of rheumatoid factor or other forms of arthritis. Plain radiography showed hazy soft tissue swelling and marked narrowing of the joint space with osteophytes, multiple tiny subchondral cysts, and bony erosion due to extrinsic compression, indicating severe osteoarthritic changes in the elbow joint, with possible soft tissue mass externally eroding the olecranon. Plain radiograph shows moderate narrowing of the joint space with osteophytes (red arrow) and multiple tiny subchondral cysts (white arrow) as well as bony erosion caused by extrinsic compression (yellow arrow), suggesting marked degenerative osteoarthritis of the elbow joint associated with probable longstanding soft tissue mass. (**b**) Follow-up radiograph four years after surgical management reveals a more stabilized joint with the disappearance of osteophytes (red arrow) and tiny cysts (yellow arrow). Clinically, the range of elbow motion, however, did not improve.

**Figure 2 diagnostics-15-01888-f002:**
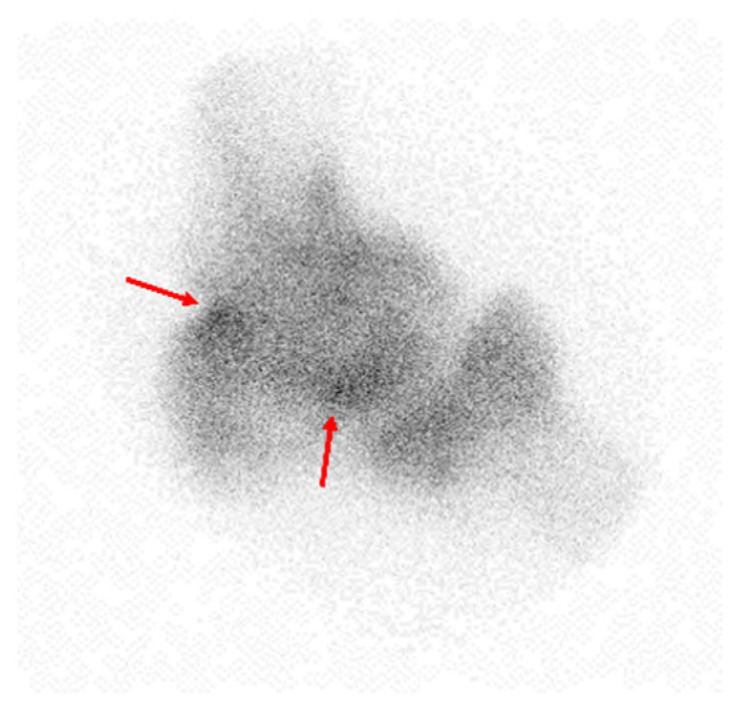
Pinhole bone scan reveals heterogeneously mild uptake with focal increased uptake (red arrows) in the distal humerus, radial head, and olecranon.

**Figure 3 diagnostics-15-01888-f003:**
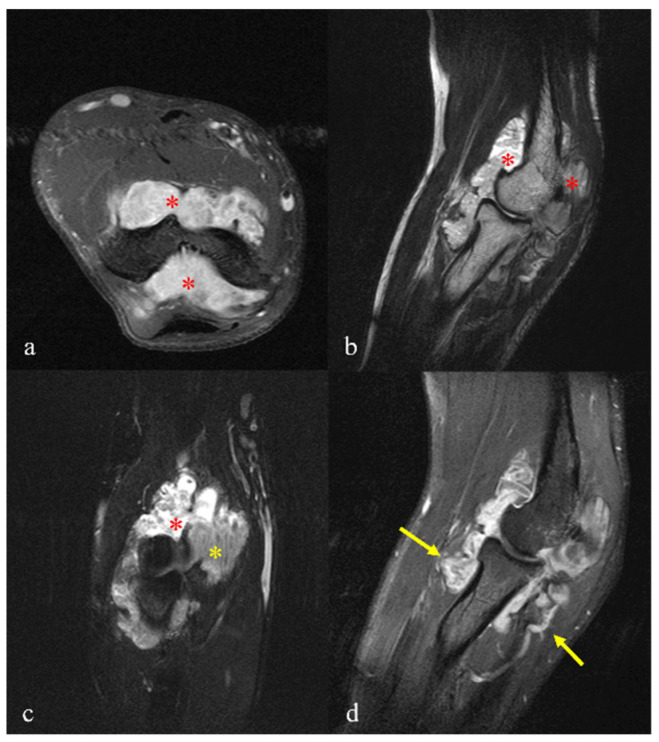
Magnetic resonance imaging (MRI) revealed an irregular, large, soft-tissue, mass-like lesion with significant synovial proliferation containing a fat component. The lesion appeared to originate from the elbow joint with profuse effusion. The synovial proliferation demonstrated a “tree-like” to “frond-like” growth pattern. The lesion markedly distended the joint capsule, with associated effusion in both the anterior and posterior aspects of the joint. The SI appears heterogeneously intermediate and high (red asterisks) on both the T1-weighed image (**a**) and T2-weighted image (**b**), being identical to that of subcutaneous fat. A coronal T2 fat-saturated image reveals areas of suppressed fat signal (yellow asterisk) with remaining areas of bright signal (red asterisk), suggesting inflammatory synovitis with effusion or edema (**c**). On the sagittal gadolinium-enhanced image, the lesion itself is not enhanced; however, the synovial membranes are peripherally and focally enhanced (yellow arrows) (**d**).

**Figure 4 diagnostics-15-01888-f004:**
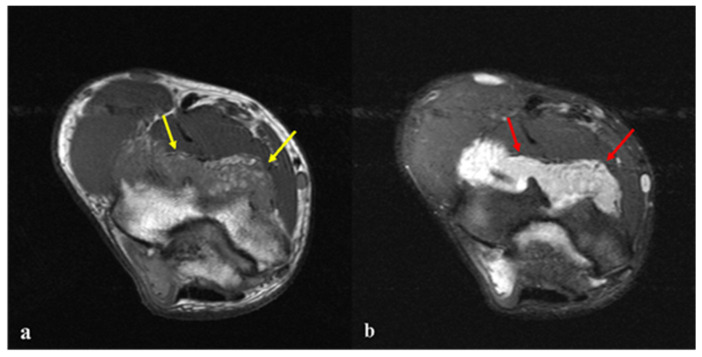
(**a**) Axial T1-weighted non-fat-suppressed image showed a heterogeneous mass with isointense to hypointense lesion (yellow arrows) within the posterior elbow joint, consistent with fatty tissue. (**b**) Axial fat-suppressed T2-weighted image revealed hyperintense lesion (red arrows), confirming its fatty composition. No internal septations, frond-like architecture, or adjacent bone erosion were present, supporting the diagnosis of a synovial lipoma. Imaging studies indicated monoarticular LA with significant degenerative osteoarthritis of the elbow joint. Lipoma arborescens (LA) is a chronically progressive, non-neoplastic, benign lesion of the synovial lining of joints or bursae, characterized by “frond-like” or “tree-like” proliferations of fatty tissue within the hyperplastic synovium [1]. The knee joint is by far the most frequently affected site, and most cases are unilateral [2,3,4,5]. However, it has rarely been reported in other joints or bursae, including the shoulder [6,7,8,9,10,11], elbow [12,13,14,15], wrist [16], hip [17,18,19,20,21], and even the ankle [22]. LA is primarily a disease of adults, typically occurring in individuals between the fourth and seventh decades; however, it has also been diagnosed in younger patients [3,8,22,23,24]. Here, we describe a unique case of secondary LA possibly contributing to severe osteoarthritic changes in the elbow with limited joint motion in a young patient.

**Figure 5 diagnostics-15-01888-f005:**
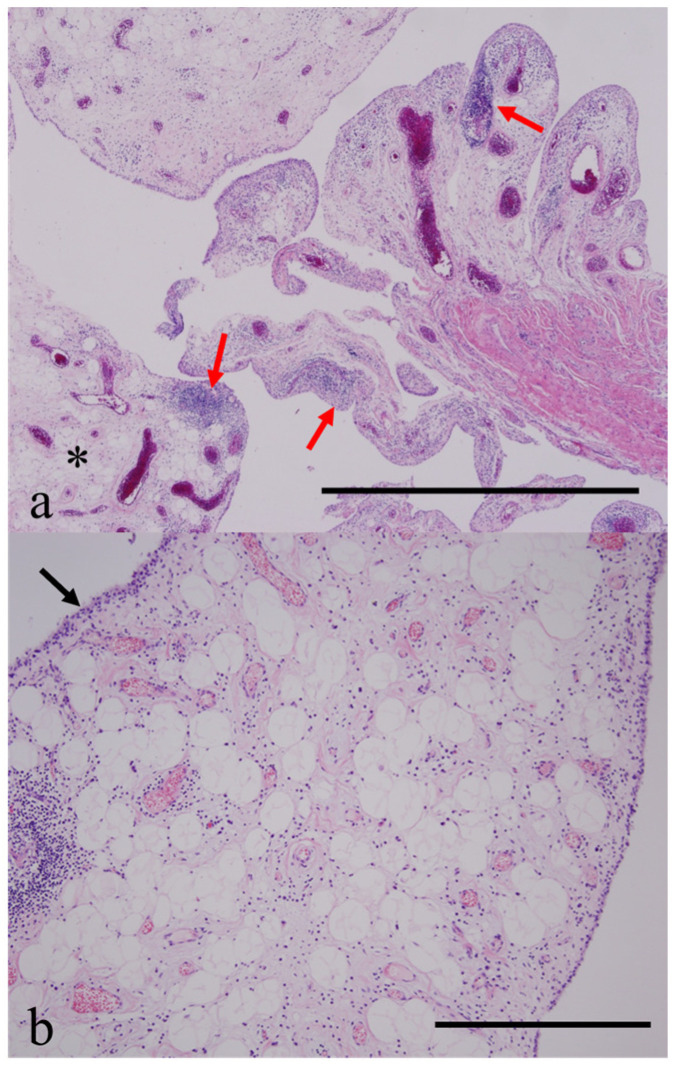
Excision of the mass lesion and synovectomy were performed. (**a**) Low-power microscopic examination (×12.5, scale bar = 5 mm) revealed multiple variable-sized villi with well-vascularized mature adipocytes (asterisks) variably expanding the villous cores. Histological examination of the resected mass and synovium showed papillary proliferation of the synovial villi with abundant, well-vascularized, mature adipose tissue, corresponding to the non-enhancing lesion seen on MRI. Dense and focally nodular lymphoplasmacytic infiltrates are present within the distended and hypertrophied synovium (red arrows), indicating coexisting inflammatory synovitis along with fat proliferation, correlating with the peripherally and focally enhancing areas on MRI. (**b**) High-power microscopic examination (×40, scale bar = 1 mm) demonstrates an elongated synovial fold markedly distended by abundant deposition of uniformly benign mature adipocytes lined by synovial cells (black arrow), consistent with lipoma arborescens (LA). Based on these findings, a final diagnosis of LA with severe osteoarthritis was established. Clinically, the patient was free of pain, though no improvement was observed in the range of elbow motion. Follow-up radiography demonstrated partial regression and stabilization of the degenerative changes four years after surgical management (Figure 1b). In the 19th century, Müller described some cases of knee joint dysfunction caused by adipose tissue. In 1904, Hoffa first used the term LA to describe the inflammatory fibrous hyperplasia of the articular adipose tissue in 21 cases, which was thought to be an important factor in producing knee joint disturbances [25]. Decades later in 1957, Arzimanoglu fully described a case affecting both knees in a 19-year-old woman. LA can be subclassified into primary and secondary types depending on the age of onset and underlying causes [4,24,26]. “Primary” LA develops de novo and is considered idiopathic with no clear underlying cause and is less common. This subtype is mainly observed in younger populations, usually between the ages of 20 and 30 years [22,24]. “Secondary” LA, also known as synovial lipomatosis, is associated with chronic underlying conditions such as degenerative disease [1,2,6,8,27,28,29,30,31,32], rheumatoid arthritis (RA) [4,31,32], sarcoidosis, or psoriatic arthritis [33], as well as trauma/injury [34] or foreign body reactions [35]. Secondary LA is much more prevalent and typically develops in elderly patients [4,24,27]. The lesion is usually monoarticular, but biarticular and multiarticular cases have been reported [11,17,23,32,36,37,38,39]. Although the exact cause of LA remains unknown, it is often found to be associated with degenerative or rheumatoid arthritis or traumatic conditions that irritate the synovium. In a large series, the majority of elderly patients (28 of 32 patients; 87%) had degenerative changes, including joint effusion, meniscal tear, synovial cysts, bone erosions, chondromatosis, patellar subluxation, and discoid meniscus; only two patients (6%) had no such changes, and both were young adults [4]. Although it has been suggested that synovial proliferation with deposition of mature fatty tissue is a non-specific response to the aforementioned inflammatory, traumatic, or mechanical irritations, the exact mechanism of LA remains unclear. LA is rare, representing less than 1% of all lipomatous lesions [40]. Since Hoffa described 21 cases, a few larger series have been published, including 45 lesions in 39 patients by Howe and Wenger and 33 lesions in 32 patients by Vilanova et al., along with several smaller series [17,37,41,42,43,44,45,46]. However, most lesions have been described in single case reports. Patients typically present with swelling with or without mild pain, of long-standing duration, often persisting several years—up to 30 years in some cases [42,47]. Occasionally, a large amount of effusion can accumulate in the affected joint, leading to a decreased range of motion. LA can also cause joint clicking [48], and a soft tissue mass may be palpable [49]. Range of motion can be markedly limited if significant osteoarthritis is present, as in our case. In a review of 39 reported cases, osteoarthritic changes were observed in 22 of 36 knee cases (61%), including 3 cases of severe osteoarthritis, suggesting LA as a possible cause of osteoarthrosis [2]. Plain radiographs usually reveal normal architecture and alignment with or without nonspecific soft tissue opacification and a large effusion surrounding the involved joint. Occasionally, imaging may reveal radiolucent areas within a soft tissue lesion, suggesting a fatty component, and often demonstrates mild to marked narrowing of the joint space, along with osteophytes, indicating degenerative osteoarthritis. Osseous changes—such as articular erosion, sclerotic changes, and subchondral cysts—are generally very rare, and these findings may be useful in differentiating LA from other joint diseases, such as pigmented villonodular synovitis and gouty arthritis [1]. In the present case, radiographs revealed marked joint space narrowing, osteophytes, and multiple subchondral cysts, indicating severe osteoarthritis. However, the patient’s negative past history and family history, as well as negative laboratory findings for other osteoarthritis such as RA or gouty arthritis, strongly suggested that the osteoarthritic changes were a reaction to a longstanding pre-existing condition, considering his young age of 24 years. In addition to LA, synovial arthritis of the elbow joint may result from a range of etiologies, including rheumatoid arthritis, osteoarthritis, and traumatic injury. Other contributing factors may include repetitive strain, overuse, and certain infectious processes. A comprehensive diagnostic approach is essential for accurate assessment and typically begins with a detailed patient history and physical examination. Key laboratory investigations often include rheumatoid factors, anti-cyclic citrullinated peptide antibodies, antinuclear antibodies, and synovial fluid analyses for crystal identification and microbiological cultures. Imaging modalities, such as plain radiography and MRI, play a critical role in characterizing joint and soft tissue changes. Although bone scintigraphy offers insight into bone metabolic activity, it is generally not considered a first-line diagnostic tool in the evaluation of arthritis due to its lack of specificity for direct joint pathology. Instead, its role is typically supplementary, particularly in cases involving unclear or widespread symptoms. However, it occasionally reveals heterogeneous mild to moderate uptake in the presence of osteoarthritic changes, as in our case, and can be useful for detecting additional lesions in other joints if existing. The diagnostic algorithm for joint arthritis generally follows a structured, stepwise approach: (1) comprehensive clinical history and physical examination; (2) classification of the joint involvement pattern (monoarthritis, oligoarthritis, or polyarthritis); (3) differentiation between inflammatory and non-inflammatory processes; (4) targeted laboratory testing; (5) joint aspiration for synovial fluid analysis when effusion is present; (6) appropriate imaging studies; and (7) advanced investigations, such as synovial biopsy or referral to rheumatology, when clinically indicated. MRI is the imaging modality of choice for diagnosing LA and typically demonstrates fat-containing, “frond-like” synovial projections or a mass-like lesion, with concurrent effusion and inflammatory changes. MRI is particularly valuable for its sensitivity in detecting early joint pathology and for differentiating between various forms of arthritis. Additionally, it provides superior visualization of soft tissue structures; however, its high cost and limited utility in systemic evaluation are recognized limitations. In a large series, the lesion was identified as having a diffuse proliferation pattern in 79% (26/33) of cases and a dominant mass-like lesion in 21% (7/33). Degenerative changes were associated in 87%, and meniscal tears in 72% of patients [4]. The SI was heterogeneously intermediate to high on both the T1- and T2-weighted images, being isointense with subcutaneous fat, depending on the degree of fat deposition. The high to bright fat signal is well suppressed on short tau inversion recovery (STIR) or fat-saturated images. The non-fat component of the hypertrophied synovium with inflammatory reaction shows an intermediate SI on T1-weighted images and heterogeneously high SI on T2-weighted or STIR sequences. The hypertrophied sub-synovial fatty tissue itself is not enhanced, but the chronically inflamed areas demonstrate diffuse enhancement [17,46]. Synovial thickening and effusion caused by chronic irritation of the proliferated villi may eventually lead to osteoarthritis [42]. LA is almost always associated with other chronic joint pathologies, particularly in elderly individuals [4]. Synovectomy is the definitive treatment, and postoperative recurrence is rare. However, primary LA may lead to early osteoarthritis if prompt synovectomy is not performed [44] and delayed surgical management may fail to improve joint mobility [2]. Patients treated with synovectomy experienced complete healing of the synovial lesion, but their osteoarthritis tended to progress—especially in those with longstanding symptoms. Thus, LA must be diagnosed early, and progressive or symptomatic lesions should be resected promptly by open or arthroscopic synovectomy to prevent osteoarthritic changes [42]. In summary, a severe degree of osteoarthritis in the absence of prior inflammatory disease or trauma in our young patient likely resulted from untreated, longstanding LA. The limitation in elbow motion did not improve due to delayed surgical treatment. Based on the clinical course and radiologic features, it is strongly suggested that unrecognized LA can chronically irritate the joint and result in significant osteoarthritis. Therefore, early recognition and timely surgical management of LA are warranted to prevent osteoarthritic changes and to preserve joint function.

## Data Availability

The data presented in this study are available upon request from the corresponding author.

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
