# Peer review of "A Lipoma Arborescens Probably Causing Significant Osteoarthritis of the Elbow in a Young Man"

_diagnostics, 2025, doi:10.3390/diagnostics15151888_

Round 1

Reviewer 1 Report

Comments and Suggestions for Authors

This case report describes a rare presentation of lipoma arborescens (LA) in the elbow joint of a 24-year-old male, associated with advanced osteoarthritic changes and significant joint dysfunction. The case is well-documented, and the authors provide a comprehensive overview of the clinical presentation, imaging findings, histopathological correlation, and management. The manuscript offers valuable insights into a condition rarely observed in this anatomical location, especially in young patients.

Some comments for contributing to improve the manuscript are:

  1. The discussion section contains repetitive content and could be more concise.
  2. A discussion on differential diagnoses for elbow joint synovial pathologies is lacking. Including a diagnostic algorithm or flowchart—particularly addressing the role and limitations of imaging modalities such as MRI and bone scintigraphy—would be beneficial. The manuscript suggests that bone scintigraphy was not contributory; this point should be clarified and contextualized.
  3. The manuscript would benefit from careful language editing. 

Conclusion:
This manuscript describes a rare and clinically relevant case of elbow LA in a young adult and has the potential to contribute meaningfully to the literature on synovial proliferative disorders. With minor revisions the report would be a valuable addition to the journal

Comments on the Quality of English Language

Must be improved.

Author Response

Authors' Responses to Reviewer's Comments (Reviewer 1)

Dear Reviewer,

Thank you for your meticulous review of our manuscript. We truly appreciate your valuable comments. The manuscript has been revised and appropriate changes have been made based on your insightful and helpful comments. The following text represents point-by-point responses to the comments. Thank you again for your consideration and excellent feedback. Your judicious comments have helped shape our manuscript into a better, more coherent version.

Point-by-Point Responses to Reviewer 1's Comments

  1. The discussion section contains repetitive content and could be more concise.

Response: We have deleted the repetitive sentences. The deleted sentences are as follows: The Latin term “arbor” means tree-like structure or system resembling a tree with its branches, and “arborescens” is used to describe the growth pattern of a lesion branching like a tree. LA is a rare, non-neoplastic joint lesion characterized by diffu se proliferation of the synovial villi with subsequent subsynovial fatty replacement appearing in a “tree-like” pattern, most commonly in the knee joint. The vast majority of LA cases have been reported in the unilateral knee joint, accounting for over 90% of cases in a large series The age at presentation varied widely from 9 to 90 years, with a mean of 40.8 years. Similar age ranges (9 to 66 years) have been reported in other studies (lines 145 to 152)

  1. A discussion on differential diagnoses for elbow joint synovial pathologies is lacking. Including a diagnostic algorithm or flowchart—particularly addressing the role and limitations of imaging modalities such as MRI and bone scintigraphy—would be beneficial. The manuscript suggests that bone scintigraphy was not contributory; this point should be clarified and contextualized.

Response: We have added a discussion on differential diagnoses for elbow joint synovial pathologies with a diagnostic algorithm, as you suggested (line 134 ~159)

  1. The manuscript would benefit from careful language editing.

Response: We have edited the English.

Reviewer 2 Report

Comments and Suggestions for Authors

I always thought that this might occur in lipoma arborecens but never so a confirming paper.  so for me it is a very interesting manuscript

Author Response

Authors' Responses to Reviewer's Comments (Reviewer 2)

Dear Reviewer,

Thank you for your meticulous review of our manuscript. We truly appreciate your valuable comments. The manuscript has been revised and appropriate changes have been made based on your insightful and helpful comments. The following text represents point-by-point responses to the comments. Thank you again for your consideration and excellent feedback. Your judicious comments have helped shape our manuscript into a better, more coherent version.

Point-by-Point Responses to Reviewer 2's Comments

In conclusion, the manuscript contributes to the medical literature by presenting a rare case of LA in the elbow, reinforcing the importance of early diagnosis and treatment to prevent osteoarthritis. Its scientific merit is evident in its detailed presentation and educational content, though its impact is tempered by its single-case nature. Prior studies provide a robust context, supporting the manuscript's findings and highlighting areas for future research, such as long-term outcomes and comparative analyses across cases.

Response: Thank you for your review of our manuscript. We truly appreciate your valuable comments. We have added some recommended references.

Reviewer 3 Report

Comments and Suggestions for Authors

A very well-presented case, with a good overview of the literature.

My advice is to shorten (in order to simplify) lines 23 to 41, in order to avoid repetition with the end of the discussion.

For the rest, the manuscript is perfect to me.

Author Response

Authors' Responses to Reviewer's Comments (Reviewer 3)

Dear Reviewer,

Thank you for your constructive review of our manuscript. We truly appreciate your valuable comments. The manuscript has been revised and appropriate changes have been made based on your comments. The following text represents point-by-point responses to the comments. Thank you again for your consideration and excellent feedback.

Your comments have helped shape our manuscript into a better, more coherent version.

Point-by-Point Responses to Reviewer 3's Comments

My advice is to shorten (in order to simplify) lines 23 to 41, in order to avoid repetition with the end of the discussion.

Response: We have reduced the text in lines 23 to 41.

(Lipoma arborescens (LA) is a chronically progressive, non-neoplastic, benign lesion of the synovial lining of joints or bursae, characterized by “frond-like” or “tree-like” proliferations of fatty tissue within the hyperplastic synovium. The knee joint is by far the most frequently affected site, and most cases are unilateral. However, it has rarely been reported in other joints or bursae, including the shoulder, elbow, wrist, hip, and even the ankle. LA is primarily a disease of adults, typically occurring in individuals between the fourth and seventh decades; however, it has also been diagnosed in younger patients. Here, we describe a unique case of secondary LA possibly contributing to severe osteoarthritic changes of the elbow with limited joint motion in a young patient.)